# Toxicity Studies of Cardiac-Targeting Peptide Reveal a Robust Safety Profile

**DOI:** 10.3390/pharmaceutics16010073

**Published:** 2024-01-04

**Authors:** Daniella A. Sahagun, Jack B. Lopuszynski, Kyle S. Feldman, Nicholas Pogodzinski, Maliha Zahid

**Affiliations:** 1Department of Cardiovascular Medicine, Mayo Clinic, Rochester, MN 55905, USA; sahagun.daniella@mayo.edu (D.A.S.); lopuszynski.jack@mayo.edu (J.B.L.); 2Clinical Virology Laboratory, Yale New Haven Hospital, New Haven, CT 06511, USA; kyle.feldman@ynhh.org; 3Department of Developmental Biology, University of Pittsburgh, Pittsburgh, PA 15260, USA; pogo812@gmail.com

**Keywords:** cell penetrating peptides, human cardiomyocyte cells, cytotoxicity, cardiac magnetic resonance

## Abstract

Targeted delivery of therapeutics specifically to cardiomyocytes would open up new frontiers for common conditions like heart failure. Our prior work using a phage display methodology identified a 12-amino-acid-long peptide that selectively targets cardiomyocytes after an intravenous injection in as little as 5 min and was hence termed a cardiac-targeting peptide (CTP: APHLSSQYSRT). CTP has been used to deliver imaging agents, small drug molecules, photosensitizing nanoparticles, exosomes, and even miRNA to cardiomyocytes. As a natural extension to the development of CTP as a clinically viable cardiac vector, we now present toxicity studies performed with the peptide. In vitro viability studies were performed in a human left ventricular myocyte cell line with 10 µM of Cyanine-5.5-labeled CTP (CTP-Cy5.5). In vitro ion channel profiles were completed for CTP followed by extensive studies in stably transfected cell lines for several GPCR-coupled receptors. Positive data for GPCR-coupled receptors were interrogated further with RT-qPCRs performed on mouse heart tissue. In vivo studies consisted of pre- and post-blood pressure monitoring acutely after a single CTP (10 mg/Kg) injection. Further in vivo toxicity studies consisted of injecting CTP (150 µg/Kg) in 60, 6-week-old, wild-type CD1, male/female mice (1:1), with cohorts of mice euthanized on days 0, 1, 2, 7, and 14 with inhalational CO_2_, followed by blood collection via cardiac puncture, complete blood count analysis, metabolic profiling, and finally, liver, renal, and thyroid studies. Lastly, mouse cardiac MRI was performed immediately before and after CTP (150 µg/Kg) injection to assess changes in cardiac size or function. Human left ventricular cardiomyocytes showed no decrease in viability after a 30 min incubation with CTP-Cy5.5. No significant activation or inhibition of any of seventy-eight protein channels was observed other than OPRM1 and COX2 at the highest tested concentration, neither of which were expressed in mouse heart tissue as assessed using RT-qPCR. CTP (10 mg/Kg) injections led to no change in blood pressure. Blood counts and chemistries showed no evidence of significant hematological, hepatic, or renal toxicities. Lastly, there was no difference in cardiac function, size, or mass acutely in response to CTP injections. Our studies with CTP showed no activation or inhibition of GPCR-associated receptors in vitro. We found no signals indicative of toxicity in vivo. Most importantly, cardiac functions remained unchanged acutely in response to CTP uptake. Further studies using good laboratory practices are needed with prolonged, chronic administration of CTP conjugated to a specific cargo of choice before human studies can be contemplated.

## 1. Introduction

Advances in the development of novel technologies in pharmaceutics today necessitate the development of safe and effective targeted delivery methods for cargoes such as nanoparticles, siRNA, drugs, and more. Cell-penetrating peptides (CPPs), which are small, positively charged peptides capable of crossing the cell membrane barrier, provide a promising method for intracellular delivery, which is highly efficient and minimally cytotoxic [1,2,3]. Tat (HIV trans-activator of transcription) was the first CPP to be discovered to enter the cell and induce the expression of viral genes [4,5], but given its nature to ubiquitously target all cell types, and even cross the blood–brain barrier, the quest to discover different cell-specific CPPs has become a major endeavor. Among other strategies, phage display libraries have been used successfully to identify peptide sequences which have cell-penetrating properties [5,6,7,8,9], leading to the discovery of many cell-specific CPPs over time [10,11]. Such CPPs could be utilized for targeted delivery of both diagnostics as well as therapeutics [12,13,14]. Our previous work with phage display led to the identification of one such CPP, which we termed a cardiac-targeting peptide (CTP: APHLSSQYSRT), due to its ability to transduce cardiomyocytes in vivo in as little as 15 min after a peripheral intravenous injection [6,15].

Heart disease and cardiac illness continue to be a leading cause of death in the US [16], and, as such, novel treatments for increasingly prevalent conditions such as heart failure and atrial fibrillation are sorely needed. CTP has shown promise as a vector for both therapeutic and diagnostic agents [17] and is capable of penetrating cardiomyocytes both in vivo in mice and in vitro in human cardiomyocytes (hCMCs) [6], exhibiting cardiac-specific transduction which is seemingly not species-limited, as it has also shown efficacy in transducing both sheep and rat models [18], as well as induced pluripotent human cells differentiated into beating cardiomyocytes [15]. CTP appears to primarily transduce the heart and is not taken up by the lungs, spleen, or other muscle subtypes, and does not cross the blood–brain barrier [15]. The cell-specific transduction ability of CTP has been validated several times by independent laboratories as being specific to cardiomyocytes and has been shown to be effective at delivering a variety of cargoes, including photosensitizers [19], exosomes [18], miRNA for the treatment of heart failure [20], and amiodarone [21] at significantly reduced doses, suggesting a potential for reduction in off-target side effects while retaining amiodarone functionality [22]. A recent publication summarized all of this accumulated literature on CTP [13].

In conclusion, CTP appears to be a novel non-viral vector, specifically targeting cardiomyocytes, and has shown promise as an innovative technology that can be utilized to deliver a myriad of different cargos. While CTP has shown no outwardly toxic effects, the authors present here systematic studies showing CTP’s lack of toxicity both in vitro and in vivo. We first present in vitro studies in stably transformed cell lines showing no effects of CTP on important receptors like calcium, sodium, potassium, or HERG2 receptors. In vivo studies show a lack of overt hepatic or renal toxicity with normal biomarkers, and no acute effect of intravenous delivery on cardiac function in mice as assessed by cardiac MRI studies. Hence, CTP shows great promise as a cardiomyocyte specific, non-toxic CPP, opening the doors to targeted delivery of established or novel therapies, the development of which has been hindered by a lack of delivery to the heart.

## 2. Methods

### 2.1. In Vitro Studies

Fluorescence-Activated Cell Sorting Assays: CTP was tested in a human left ventricular myocyte cell line (Celprogen, Torrance, CA, USA; cat #36044-15) for uptake and assessment of its effects on cell viability. Cells were plated onto 6-well plates at a density of 100,000 cells per well and incubated overnight at 37 °C/5% CO_2_. The following day media was replaced with fresh media containing 10 µM CTP-Cy5.5, or a random (RAN-CY5.5) peptide. A yellow live-dead stain (Invitrogen, Carlsbad, CA, USA; cat #L34968) was added to all wells, including the negative control or vehicle only treated controls, and cells were incubated for 30 min, after which media aspirated, cells washed 3× with pre-warmed phosphate-buffered saline (PBS), trypsinized, and collected. After collection, cells were washed once with PBS, fixed with 2% paraformaldehyde at room temperature for 10 min, washed once again and resuspended in 1000 µL of PBS. Fluorescence activated cell sorting (FACS) was performed using lasers selected for live-dead and Cy5.5 fluorophores, and 10,000 cells counted in Fortessa. The data generated were gated on the presence of the live-dead stain providing the percent viability of cells [(live/(live + dead cells)) × 100] after treatment with various peptides. 

Eurofins Toxicology Study: In vitro toxicology studies were performed by Eurofins Scientific through both their Ion Channel Profiler and Cardiac Profiler Panel and their DiscoverX’s SAFETYscan E/IC50 SELECT service. Increasing concentrations of CTP (0.1 µM to 30 µM) were tested against various ion channels and a total of seventy-eight assays performed utilizing various output reads, including GPCR cAMP modulation assays, calcium mobilization assays, nuclear hormone receptor assays, KINOMEscan binding assays, ion channel assays, transporter assays, and enzymatic assays. All methods described below were performed, described, and reported by Eurofins Scientific in their study report (Appendix A). 

#### 2.1.1. Eurofins Ion Channel Cardiac Profiler Panel

Detailed methods can be found in Appendix A. Electrophysiological assays were conducted on four types of channels: voltage-gated sodium channels HEK-Nav1.5 (peak) and HEK-Nav1.5 (late, antagonist); voltage-gated potassium channels HEK-Kv4.3/KChIP2, CHO-hERG, and CHO-KCNQ1/minK; voltage-gated calcium channel HEK-Cav1.2; and inward-rectifying voltage gated potassium channel HEK-Kir2.1. IC_50_ values following CTP treatment at concentrations ranging from 0.1 to 30 µM were calculated using non-linear, least squares regression analyses, and compared to reference standards. 

#### 2.1.2. Eurofins SAFETYscan E/IC50 ELECT Service

GPCR cAMP modulation assays, calcium mobilization assays, nuclear hormone receptor assays, KINOMEscan binding assays, ion channel assays, transporter assays, and enzymatic assays were performed by Eurofins under GLP standards. Simplified methods are provided below; detailed methods can be found in Appendix A. 

GPCR cAMP Modulation Assays: cAMP Hunter cell lines were grown, plated, and subjected to Eurofins’ DiscoverX HitHunter cAMP XS+ assay. Gs agonist activity, Gi agonist activity, and antagonist activity-induced assays were read using chemiluminescent signal detection. The percentage of activity was calculated for Gs and Gi agonist assays, and the percentage of inhibition was calculated for Gs antagonist assays.

Calcium Mobilization Assays: PathHunter cell lines were grown, plated, and subjected to calcium mobilization assays using the DiscoverX Calcium No Wash^PLUS^ kit. Agonist and antagonist activity of CTP was performed and measured on a FLIPR Tetra using a five-second baseline read followed by 2 min of calcium mobilization monitoring. The area under the FLIPR curve and percentage activity was calculated to determine agonist activity, and percentage inhibition was calculated to determine antagonist activity. 

Nuclear Hormone Receptor Assays: PathHunter NHR cell lines were grown, plated, and subjected to nuclear hormone receptor assays using the PathHunter Detection reagent cocktail. Chemiluminescent signals were read using a PerkinElmer Envision instrument, and percent activation and percent inhibition were calculated to determine agonist and antagonist activity, respectively. 

KINOMEscan Binding Assays: Kinases were produced by the infection of BL21 *E. coli* using kinase tagged T7 phage strains. Cells were grown, lysed, centrifuged, filtered, and the remaining kinases produced in HEK-293 cells were tagged using DNA for detection using qPCR. Affinity resins for the kinase assays were created using streptavidin-coated magnetic beads treated with biotinylated small molecule ligands. Kinases, beads, and CTP were combined in 1x binding buffer to perform the assay. Kinase concentrations were measured using qPCR, and the percent of response and binding constants were calculated. 

Ion Channel Assays: Cell lines were grown, plated, and loaded with 1× loading buffer containing 1× dye and 2.5 mM freshly prepared probenecid. Assays were performed, and activity was measured using a FLIPR Tetra. Percent of activation and percent of inhibition were calculated to determine agonist and antagonist activity, respectively.

Transporter Assays: Cell lines were grown, plated, and incubated first with CTP, then with 1x loading buffer containing 1× dye with 1× HBSS and 20 mM HEPES. Fluorescence signals were detected using a PerkinElmer Envision, and the percentage of inhibition assessed. 

Enzymatic Assays: AChE, COX1, COX2, MAOA, PDE3A, and PDE4D2 activity assays were performed to determine CTP effects on each individual enzyme. Fluorescence was read using a PerkinElmer Envision, and percentage inhibition and compound activity calculated. 

### 2.2. In Vivo Studies

All animal protocols were approved by the institutional animal care and committee of the University of Pittsburgh and Mayo Clinic. CTP was synthesized by the University of Pittsburgh using standard solid-state synthesis, followed by HPLC purification. MALDI LC/MS characterization of the final peptide was performed prior to use. 

Blood Pressure Pre- and Post-CTP Injection: Six B6-wildtype mice were obtained from the Jackson Laboratory. Blood pressure was taken using a non-invasive blood pressure system from Kent Scientific. Mice were anesthetized using 2.5% isoflurane exposure for 5 min, placed in a tube holder, and warmed using a warming blanket for approximately 15 min before their tails were fitted with blood pressure cuffs. A total of 21 blood pressure readings were taken, and average systolic and diastolic blood pressure calculated. After baseline blood pressure was established, mice were injected intravenously with 10 mg/Kg of CTP, and blood pressure was measured again 6 h post-injection.

Mouse Toxicity Studies: Sixty CD1-wildtype mice (1:1 male to female ratio) were obtained from Charles River and divided into 6 groups of 10, 5 females and 5 males each. Ten control mice were injected with 1X PBS. Treated mice were injected intravenously with a single dose of CTP 150 μg/Kg. Groups of 10 treated mice (5 male, 5 female) were euthanized on day 0 (immediately after injection), 1, 2, 7, and 14 post-injection using inhalational CO2. After euthanasia, the chest cavity was opened, and blood drawn via cardiac puncture for complete blood count and serum chemistries. Following the blood draw, the right atrium was nicked and the whole animal perfusion fixed using 3 mL of formalin injected into the left ventricular apex. The control group was euthanized on day 2. A complete blood count, metabolic profile, and thyroid studies were performed.

Cardiac Function Studies, Pre- and Post-Injections: Six CD1, wild type, 6–8-week-old adult mice were obtained from the Charles River and used for cardiac MRI studies. Mice were weighed, anesthetized with 2.5% isoflurane, and ear tags removed before being inserted into the MRI scanner. A baseline cardiac MRI was performed, followed by intravenous CTP injection (150 µg/Kg). Post-injection, a second cardiac MRI was performed. 

Real-time quantitative PCR: Based on the results of the Eurofins Toxicology Study (Figure 1), targets for qPCR were identified for expression of the activated receptors, OPRM1, COX1, and COX2 in mouse hearts. Heart, brain, kidney, liver, and lungs were harvested from an untreated, wild-type CD1, adult mouse, and snap frozen in liquid nitrogen. RNA extraction was performed using the Qiagen RNeasy Micro Kit (Sigma-Aldrich, St. Louis, MO, USA) and cDNA libraries prepared using the high-capacity RNA to cDNA kit (Applied Biosciences, Beverly, MA, USA). RT-qPCR was performed (ABI Quant, ThermoFisher, Waltham, MA, USA) using PowerUp SYBR Green Master Mix (Applied Biosciences, Tucson, AZ, USA). Results were normalized to GAPDH expression.

## 3. Discussion

The initial report of Tat’s ability to deliver beta-galactosidase, a payload much larger than Tat, in an intact, functional form was met with a lot of excitement [23,24]. However, Tat technology did not translate into a clinical application due to the ubiquitous transduction of multiple tissue types, including neuronal tissue with crossing of the blood–brain barrier. These hurdles were initially ameliorated by topical use of Tat conjugated to a therapeutic [25,26,27], or in a compartment-restricted manner such as intra-articular [28,29,30] or intra-ocular administration [31,32,33]. Another use of non-specific CPP was the use of activatable peptides given a certain tissue milieu in oncological applications [34,35,36,37,38] or enzyme-activatable cell penetrating peptides [39,40,41]. Phage display to identify specific tumor- or tissue-targeting CPPs was yet another strategy in attempts to harness CPP’s transduction abilities for clinical application [9,42,43,44]. The road from bench to bedside is a long and arduous one with many pitfalls. The toxicity of CPPs, or the combination of a particular CPP with its cargo, is another major hurdle. 

Our work with phage display led to the identification of our 12-amino-acid-long cardiomyocyte-targeting peptide, CTP. It is a non-naturally occurring peptide with little homology to any naturally occurring peptides/proteins, and unlike non-cell-specific CPPs, it is not composed of predominantly cationic amino acids. Our previous work has shown that it is taken up by cardiomyocytes in as little as 15 min after a peripheral intravenous injection [6,15]. Our studies and multiple investigations have shown that it targets cardiomyocytes specifically while sparing the other cells (fibroblasts, endothelial cells, myofibroblasts) in the heart [19]. To further CTP along in its journey towards becoming a clinically useful cardiomyocyte-specific vector, we performed the current set of in vitro and in vivo toxicity studies. 

In the current body of work, we show that CTP did not activate or inhibit multiple different key channels like HERG2, sodium, potassium, or calcium channels in which cardiomyocytes are particularly rich. Activation or inhibition could lead to significant arrhythmias, an unacceptable and potentially life-threatening toxicity. We also show that CTP did not change the cell viability of a human myocyte cell line. For our in vivo toxicities, using a non-trivial dose of CTP (10 mg/Kg), we show no change in blood pressures. Cardiac function assessed by the most reliable imaging method available (cardiac MRI) showed no significant difference in cardiac function either in immediate post-treatment images. We also performed toxicity studies after a single dose in keeping with timelines and doses suggested by the FDA for “microdose” imaging studies. We chose this path to take, as CTP is platform technology that can have both diagnostic and therapeutic applications. Using peptides for imaging applications involves a single dose, to be limited to no more than 100 µg in a 70 Kg human (approx. 1.4 µg/Kg), with toxicities performed at 100× the maximum allowed dose. This led to our current testing dose of 150 µg/Kg. At that dose, there were no significant toxicity signals noted except for a borderline increase in alkaline phosphatase levels in female mice on Days 0 and 1, with recovery to control levels by Day 2. This statistical significance did not reach clinical significance thresholds, where a >3× the upper limit of normal for alkaline phosphatase is considered a meaningful elevation. In female mice, the white blood cell and platelet count decreased on Day 0, likely due to volume expansion associated with the injection. Male mice showed a decrease in blood urea nitrogen and creatinine levels which goes against any nephrotoxicity being associated with CTP. 

Our results with CTP are consistent with reports in the literature on other CPPs [45,46,47]. Obviously, the toxicity profile will vary from one peptide to the next, and depend on the dose tested. However, the published literature suggests that toxicities remain low [48,49,50,51]. Saar and colleagues investigated membrane toxicity in cancer cell lines of five well-characterized, non-cell-specific CPPs: Ant, Tat, pVEC, MAP, and transportan 10, and found that the first three caused minimal membrane leakage of lactate dehydrogenase, whereas the latter two (MAP, transportan 10) caused significant leakage. However, none of the tested peptides had a hemolytic effect on bovine red blood cells. Another study evaluated the immunogenicity of transportan 10 and its chemical modified derivatives, PepFects, in a leukemia and peripheral blood monocyte cell line by evaluating release of multiple cytokines and apoptosis in response to incubation with these peptides at 5 and 10 µM concentrations and found no increase in cytokine levels [51]. Additionally, in vivo toxicity was evaluated by injecting immunocompetent mice with 5 mg/Kg of the peptides and blood collected at 24 and 48 h for ELISA-based estimation of IL-1β and TNF-α [52]. There was no enhancement of any of these cytokine levels in response to peptide injections. Changes to cell metabolism in response to 5 CPPs (transportan, Ant, Tat, nona-arginine, MAP) were tested in a Chinese hamster ovarian cell line. Analysis of the cell lysates was performed using liquid chromatography–mass spectrometry and showed the most significant changes in response to transportan, followed by Tat and MAP. Transportan affected cellular redox potential, depleted energy, and the pools of purines/pyrimidines [53]. Cells could recover from these effects at 5 µM of transportan [52], but not at higher treatment concentrations. Clearly, toxicities are going to be dependent on the type of CPP being studied and the concentrations utilized in these studies [53].

## 4. Results

### 4.1. In Vitro Studies

Fluorescence-Activated Cell Sorting (FACS): Studies were conducted in biological triplicates and technical quadruplicates (n = 4), showing that cell viability after treatment with 10 µM of CTP remained unchanged compared to no treatment (live-dead stain only) or vehicle treatment only. Random peptides did influence cell viability (reducing it by almost 50%), although these findings are irrelevant and only prove to support the non-cytotoxic nature of CTP (Figure 2). 

### 4.2. Eurofins Toxicology Study

IonChannelProfiler CardiacProfiler Panel: Across all seven ion channels tested, no inhibition/activation was observed. No significant inhibition was noted as compared to the control substances for any of the seven ion channels (Figure 3), and therefore IC_50_ values were not calculable (Appendix A).

SAFETYscan E/IC50 ELECT Service: Across all seventy-eight assays, only two showed significant interactions between CTP and the assayed protein. In ion channel assays, CTP showed no significant blocking or activation of hERG, NAV1.5, or KvLQT1/minK, nor any significant opening action for KvLQT1/minK (Figure 1). In GPCR cAMP modulation assays, CTP showed no significant action as either an agonist or antagonist of ADRA2A or ADRB1 compared to the controls (Figure 1). In similar assays performed using OPRM1, CTP did show agonist action with an EC_50_ of 24.975 μM, although no antagonist action of OPRM1 was noted (Figure 1). In the enzymatic assays performed, some inhibition of COX1 was noted, although it did not reach the set significance threshold. Significant inhibition of COX2 was noted with an IC_50_ of 16.869 μM. Results of all seventy-eight assays are shown (Appendix A). In order to study whether activation of these two receptors was clinically relevant, we performed RT-qPCR on RNA extracted from wild-type mouse hearts. Results were normalized to GAPDH expression. Neither COX2 nor OPRM1 showed significant expression in heart tissue with COX2 expressed at a high level in lung and liver, and OPRM1 expressed in brain, lung, and liver (Figure 4). 

### 4.3. In Vivo Studies

Blood Pressure Pre- and Post-Injection: No significant difference was measured in systolic or diastolic blood pressure between pre- and post-injection measurements (Figure 5). According to Janssen [54], average blood pressure values in wild-type mice range from ~90 to 115 mmHg.

Toxicology Studies: There was no significant difference in weights between the control and injected mice of each sex during the two-week course of the study (Figure 6). In male mice, there was no difference noted in hemoglobin, white blood cell or platelet count, alanine aminotransferase, aspartate aminotransferase, alkaline phosphatase, total bilirubin, or phosphorous levels. Kidney function studies in male mice (blood urea nitrogen and creatinine) were significantly lower on Days 0–2 than the control mice, ruling out overt kidney damage. Calcium levels increased marginally only on Day 0, normalizing at later time-points.

Female mice showed similar results. There was no significant difference overall between the treated groups and the control group in hemoglobin, alanine aminotransferase, aspartate aminotransferase, blood urea nitrogen, or creatinine levels. A significant decrease in bilirubin, white blood cell count, and platelet count was noted only on Day 0, which recovered by Day 1 for both. The only consistent change noted was a rise in alkaline phosphatase for female mice (Figure 6). However, despite the rising alkaline phosphatase, the mean values of alkaline phosphatase level did not cross the upper limit of normal (140 IU/L) for any of the groups, and only two mice had borderline elevated levels (151 and 166) on day 0, with all levels returning to baseline by Day 2. 

Cardiac MRI showed no significant change in heart rate, left ventricular mass, ejection fraction, or cardiac output post-CTP injection in male or female mice (Figure 7). 

## 5. Conclusions

CTP, as a single injection at least, is a safe and effective new technology for the delivery of therapeutic cargoes to cardiomyocytes of the heart. CTP showed no overt toxicities, and caused no major changes in hematological counts, blood chemistries or MRI-based cardiac function. In vitro cell assays showed an activation of OPRM1 and COX 2 receptors, neither of which is expressed in the normal murine heart or shown to express in large proteomics-based datasets of the human heart. CTP is a promising, novel drug delivery method, and studies to deliver diagnostic imaging agents and therapeutics to the heart are ongoing. 

Our study has several limitations worth noting. The studies were done in a non-GLP environment, with a single dose of CTP studied in vivo. Although we did not find any toxicities associated with CTP, the same cannot be said of CTP conjugated to a specific cargo, or repeated dose administrations. Toxicity studies with a specific payload will need to be repeated in two vertebrate animal species, at least one of which must be a non-rodent one, in a longer timeline fashion, and in a GLP environment before CTP can make it as a viable vector to its first Phase I human study. 

## Figures and Tables

**Figure 1 pharmaceutics-16-00073-f001:**
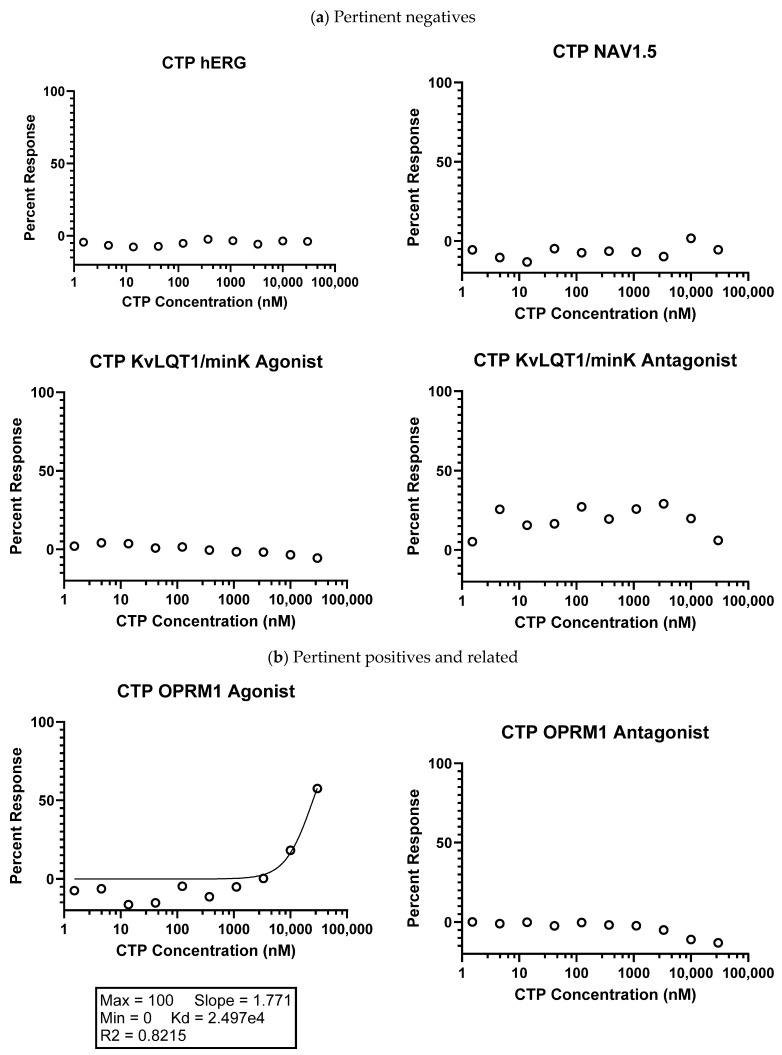
Eurofins CTP SAFETY scan results. (**a**) Highlighted pertinent negative results; (**b**) pertinent positives and related negatives; and (**c**) other negative results. Data shown were normalized to the maximal and minimal responses observed in the presence of control ligand and vehicle, respectively (y-axis), and is plotted against the corresponding compound concentration in nM in log10 scale (x-axis).

**Figure 2 pharmaceutics-16-00073-f002:**
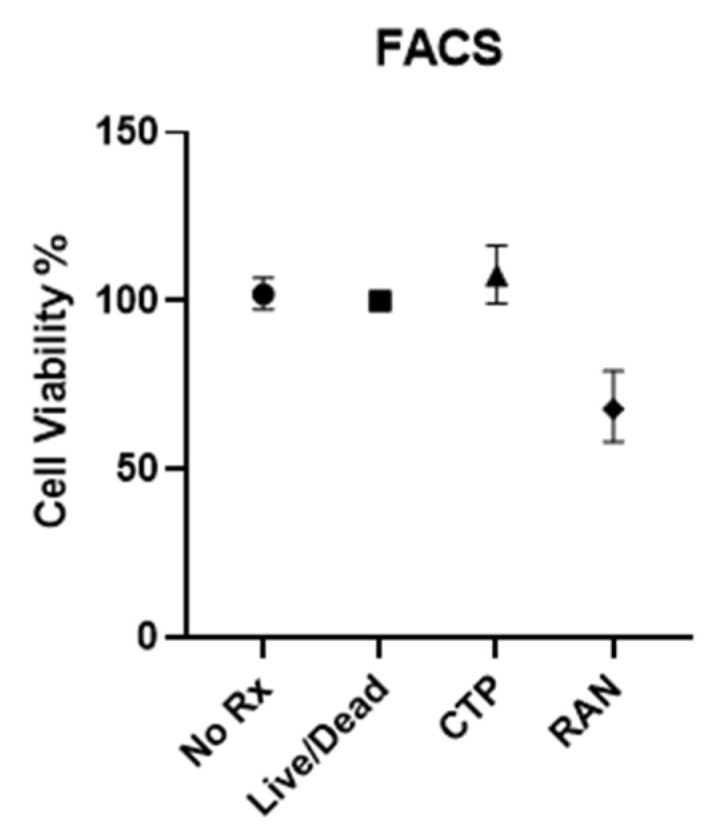
Fluorescence-activated cell sorting (FACS) assay of human ventricular myocyte cell line.

**Figure 3 pharmaceutics-16-00073-f003:**
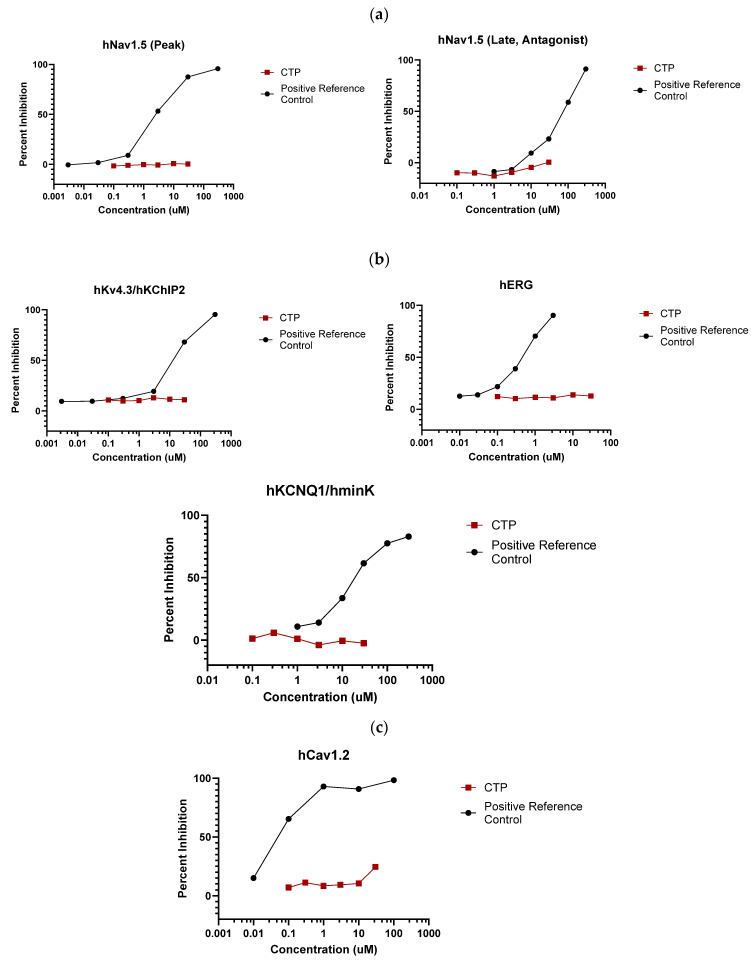
Eurofins ion channel profiler results. CTP did not inhibit the activation of any tested (**a**) voltage-gated sodium channels; (**b**) voltage-gated potassium channels; (**c**) voltage-gated calcium channels; or (**d**) inward-rectifying voltage-gated potassium channels.

**Figure 4 pharmaceutics-16-00073-f004:**
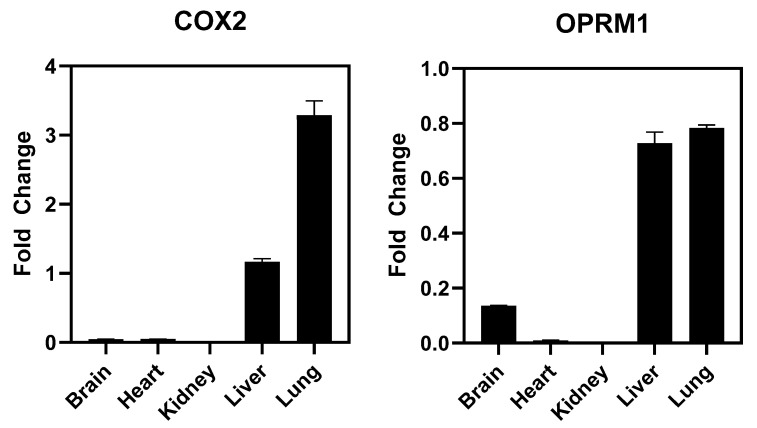
RT-qPCR Results for COX2 and OPRM1. Values are normalized to GAPDH expression.

**Figure 5 pharmaceutics-16-00073-f005:**
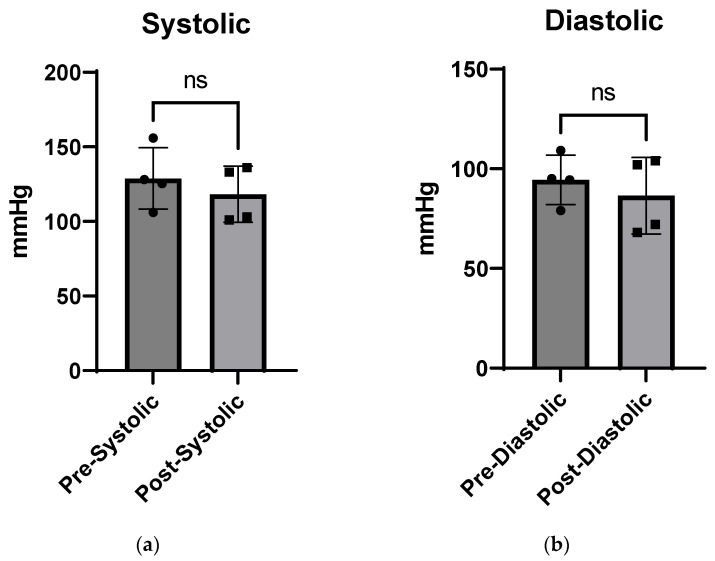
Blood pressure pre- and post-CTP injections, showing systolic (**a**) and diastolic pressure (**b**) after a single CTP injection (10 mg/Kg) with no significant (ns) change in either parameter.

**Figure 6 pharmaceutics-16-00073-f006:**
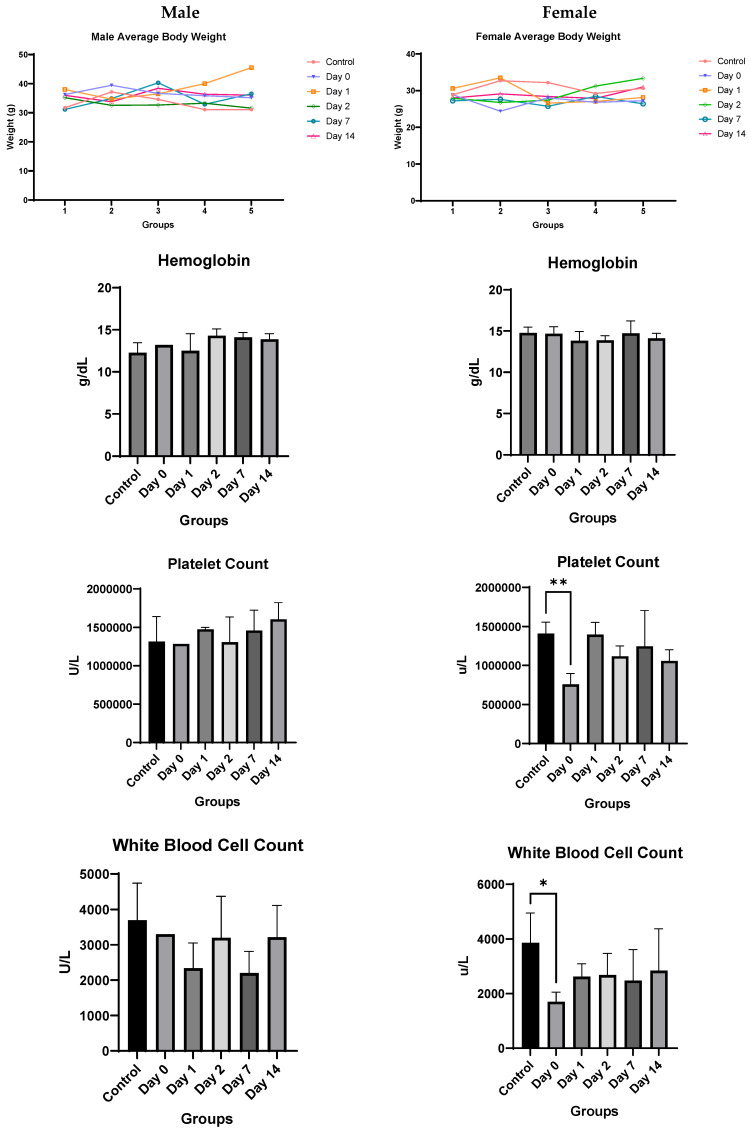
In vivo toxicity results in male and female mice. Significance markers are added and are to be interpreted as follows: *: *p* < 0.05, **: *p* < 0.01, ****: *p* < 0.0001.

**Figure 7 pharmaceutics-16-00073-f007:**
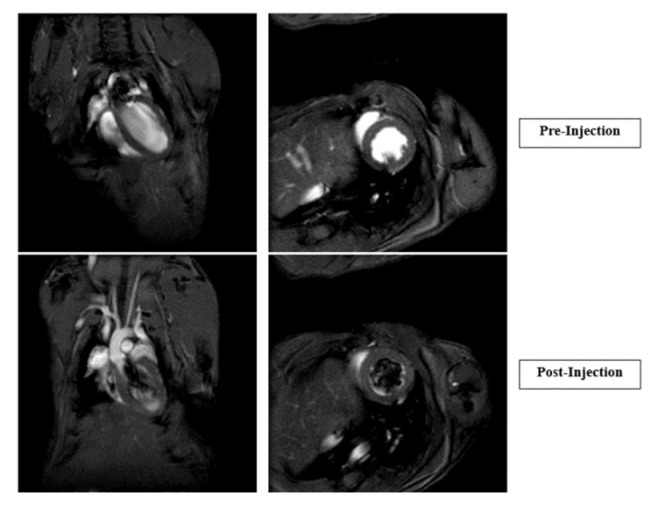
Cardiac MRIs of pre- and post-injection of CTP. Graphs showing heart rate, left ventricular (LV) wall mass, and 2D and 3D cardiac output immediately after CTP injection.

## Data Availability

The data presented in this study are available in this article and Appendix A.

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
