# Peer review of "Toxicity Studies of Cardiac-Targeting Peptide Reveal a Robust Safety Profile"

_pharmaceutics, 2024, doi:10.3390/pharmaceutics16010073_

Round 1

Reviewer 1 Report

Comments and Suggestions for Authors

The authors evaluated the security of cardiac targeting peptide (CTP), which is showed no activation or inhibition of GPCR-associated receptors in vitro, and no signals indicative of toxicity in vivo. Generally, the manuscript is hard to read for the tedious contents and the low-quality figures. 

1.Fig 1(page 3):cell viability in group CTP is nearly 50% higher than group RAN,  but the conclusion (page 2, Line 96-97) is “showing that cell viability did not change significantly after treatment with 10µM 96 of CTP compared to no treatment (live-dead stain only), vehicle, or random peptide”, please confirm the results and statements.

2.Fig 6:

(1)  The picture named “Platelet Count” (page 9) is not clear, it should be replaced by a clearer one.

(2) The picture named “White Blood Cell Count” of group “Male” (page 10) is not clear, it should be replaced by a clearer one.

(3) The meaning of symbols “*” (page 9-12) are lack of explanation in statistical differences, please add the explanation.

3. Fig 7(page 13) :the picture on the right is not clear, it should be replaced by a clearer one.

4. Page 8, Line 163-163:the sentence “There was no significant difference in weights during the two-week course of the study in male or female mice” is ambiguous, please indicate whether there is no difference between or within groups (Male & Female).

5. The abstract is tedious and unfocused.

6. Quite a lot of references are too old to support the view. 

Comments on the Quality of English Language

The language is acceptable.

Author Response

Dear Reviewer, on behalf of my co-authors I would like to thank you for your time and meticulously detailed review of our paper. We have made significant changes to the manuscript in response to your comments, and we believe this has resulted in an improved manuscript. The changes made are detailed below. 

1.Fig 1(page 3):cell viability in group CTP is nearly 50% higher than group RAN,  but the conclusion (page 2, Line 96-97) is “showing that cell viability did not change significantly after treatment with 10µM 96 of CTP compared to no treatment (live-dead stain only), vehicle, or random peptide”, please confirm the results and statements. 

We have clarified that statement. The message of the figure was that treatment with CTP did not reduce cell viability compared to vehicle only and non-treated cells. Random peptide showed some cytotoxicity, and the data is accurate.

2.Fig 6:

(1)  The picture named “Platelet Count” (page 9) is not clear, it should be replaced by a clearer one. This graph was replaced with a clearer version.

(2) The picture named “White Blood Cell Count” of group “Male” (page 10) is not clear, it should be replaced by a clearer one. Figure 7 was made clearer and larger. 

(3) The meaning of symbols “*” (page 9-12) are lack of explanation in statistical differences, please add the explanation. The meaning of the asterisks was added to the figure legend. 

3. Fig 7(page 13) :the picture on the right is not clear, it should be replaced by a clearer one. Figure 7 was made clearer and larger. 

4. Page 8, Line 163-163:the sentence “There was no significant difference in weights during the two-week course of the study in male or female mice” is ambiguous, please indicate whether there is no difference between or within groups (Male & Female).

We meant to say that there was no difference in weights of mice injected with CTP compared to control mice over time in either male or female mice. We have clarified that statement. This sentence was changed to: "There was no significant difference in weights between the control and injected mice of each sex during the two-week course of the study ."

5. The abstract is tedious and unfocused.

We have rewritten, and significantly shortened the abstract, making it more focused. 

6. Quite a lot of references are too old to support the view. 

We have added a very recent review article on CTP. However, we elected to keep the old references as we feel this orients the reader to the history of cell penetrating peptide allowing them a clearer vision of where this technology is heading. 

Reviewer 2 Report

Comments and Suggestions for Authors

The manuscript entitled "Toxicity Studies of Cardiac Targeting Peptide Reveal a Robust Safety Profile" is a third study of the Zahid group on the specific cardiac targeting peptide. Previous studies showed interesting biological activity of this peptide, therefore in the current study authors investigated its potential toxicity. The topic is original or relevant in the field. This is a well-designed study, that uses a number of biological assays to answer the main question and provides enough interesting data that warrants, in my opinion, its publication in Pharmaceutics. The obtained results show that the studied CPT should be safe and the conclusions are fully supported by the data presented. The writing is very clear and all tables and figures are informative.

That said, I would suggest at least a minor edition to make several changes. First, the abstract is definitely too long and provided too detailed information; this should be shortened by a lot and focus only on the main results. Second, captions to figures (particularly Figure 2) should just state what is presented in the figure in question, and not provide the description of results or conclusions. Figure 5 should probably have these two charts next to each other and not one below the other, to save space. Also, most of the manuscript are charts and I just wonder if there is a better way to present the large number of data in this work.

Author Response

Dear Reviewer, 

On behalf of my co-authors, thank you for your kind words of encouragement. We agree. This data is a necessary steppingstone to making the field of cell penetrating peptides a viable clinical entity. 

In response to our comments, we have shortened the abstract significantly, removed results and discussion from figure descriptions, and redone the Figure 5 formatting.

Again, thank you for your time, and input into our work.  

Reviewer 3 Report

Comments and Suggestions for Authors

The article “Toxicity Studies of Cardiac Targeting Peptide Reveal a Robust Safety Profile” by Sahagun et al. is an interesting experimental work focusing on the toxicity of targeted delivery of cell-penetrating peptides to cardiomyocytes.

The article has the following shortcomings:

1.      The keywords: The authors did not adhere to MeSH for correctly choosing the keywords. Some keywords are already in the title and therefore should NOT have been listed as keywords.

Please, see lines 44-45. Please, also correct: “Cytotoxicity” from line 44.

2.      The abstract should be rewritten more rigorously.

3.      A better representation of the study and the work done is needed for the reader to demonstrate the results. A flow diagram to describe the experiments through the various phases of this work would be most welcome.

4.      Figures 4-7 must be redesigned using a blind-color palette, so that the reader can better distinguish the variations of the variables, which can NOT be correctly identified in the shades of gray proposed by the authors.

5.      The conclusions must be reformulated more strictly.

6.      A “List of Abbreviations” must be completed and reviewed carefully and may be better presented in a table format at the end of the article.

7.      A Graphical Abstract would be very welcome to increase the impact of this article and the Pharmaceutics’ citations.

8.      A final reading on the MDPI platform by a native English speaker would be welcome.

I congratulate the authors for their work.

Overall, the study is very interesting and deserves to be published with major corrections. Overall, I recommend a major revision.

I believe that after this revision provided by the authors on the issues suggested to be corrected and improved, it will provide useful and credible information for all readers, especially researchers and clinicians, and it is up to the Academic Editor to decide on its publication.

Thank you very much!

December 2, 2023

Comments on the Quality of English Language

A final reading on the MDPI platform by a native English speaker would be welcome.

Author Response

Dear Reviewer, on behalf of my co-authors I would like to thank you for your meticulously detailed review of our manuscript. We believe that responding to those critiques has resulted in an improved manuscript. The following changes have been made:

  1. The keywords: The authors did not adhere to MeSH for correctly choosing the keywords. Some keywords are already in the title and therefore should NOT have been listed as keywords. We have rectified the mistake in our choice of key words.

Please, see lines 44-45. Please, also correct: “Cytotoxicity” from line 44.

We have corrected the mistake. 

  1. The abstract should be rewritten more rigorously. The abstract has been shortened and rewritten to make it more focused. 
  2. A better representation of the study and the work done is needed for the reader to demonstrate the results. A flow diagram to describe the experiments through the various phases of this work would be most welcome. We understand the Reviewer's concern regarding the complexity of the study as well as the number of different assays/results to report. We did divide both methods and results sections into "In vitro" and "In vivo" studies. As they are separate assays and not necessarily related, and due to the number of graphs/plots already presented, we are afraid that adding yet another diagram would muddy the waters further. 
  3. Figures 4-7 must be redesigned using a blind-color palette, so that the reader can better distinguish the variations of the variables, which can NOT be correctly identified in the shades of gray proposed by the authors. We have reviewed the identified figures and have provided labels stating the variable beneath each individual bar, making it identifiable by the label underneath it, obviating the need for color coding. 
  4. The conclusions must be reformulated more strictly. We have tempered our conclusions. The limitations follow the conclusions immediately to alert the reader to the limitations of the study conclusions.
  5. A “List of Abbreviations” must be completed and reviewed carefully and may be better presented in a table format at the end of the article. We have added a list of abbreviations on page 21.
  6. A Graphical Abstract would be very welcome to increase the impact of this article and the Pharmaceutics’ citations. As this is a number of separate assays and experimental protocols going into one study, a graphical abstract would be very complicated, and not beneficial, in this study. 
  7. A final reading on the MDPI platform by a native English speaker would be welcome. We have had two native English speakers review our writing. 

Reviewer 4 Report

Comments and Suggestions for Authors

The communication entitled 'Toxicity Studies of Cardiac Targeting Peptide Reveal a Robust Safety Profile' explores the safety of delivering therapeutic cargoes to cardiomyocytes to the heart. These studies are intriguing, and their conclusions are supported by the results. It would be valuable if the authors could provide a direct comparison of their results with those of other similar studies published in the literature. Additionally, the abstract should be shortened and Figure 6 should be replaced with a figure containing only pertinent information for the discussion. The bar charts in Figure 7 should be made clearer by increasing their size.

Author Response

Dear Reviewer, we thank you for your time and input into our manuscript. In response to your, and other reviewer's comments, we have rewritten the abstract, shortening it significantly-we believe it is more focused now, as well as increased the size of the bars in Figure 7 to make it easier to read. Please note that each bar is an individual mouse with pre- and post-injection imaging parameters displayed. Lastly, we kept the panels in Figure 6 as that is the gist of the paper and provides positives and pertinent negatives by gender.  

Round 2

Reviewer 1 Report

Comments and Suggestions for Authors

The authors have addressed all of my concerns.

Reviewer 2 Report

Comments and Suggestions for Authors

Authors of the corrected manuscript "Toxicity Studies of Cardiac Targeting Peptide Reveal a Robust Safety Profile" have addressed all of my concerns, so in my opinion the study is ready to be published.

Reviewer 3 Report

Comments and Suggestions for Authors

Since the authors responded positively to the revision prompts, I believe the manuscript has improved. However, they did not want to design a graphical abstract, arguing that it would be too difficult!

It is up to the Academic Editor to decide whether to publish the paper in the present form. Thank you very much!

December 8, 2023